# Fungal oxylipins direct programmed developmental switches in filamentous fungi

Mengyao Niu[1], Breanne N. Steffan[1], Gregory J. Fischer[2], Nandhitha Venkatesh[3], Nicholas L. Raffa[1], Molly A. Wettstein[1], Jin Woo Bok[1], Claudio Greco [1], Can Zhao[4], Erwin Berthier[1,5], Ernst Oliw[6], David Beebe [5,7], Michael Bromley [4] & Nancy P. Keller [1,8 ✉]

Filamentous fungi differentiate along complex developmental programs directed by abiotic and biotic signals. Currently, intrinsic signals that govern fungal development remain largely unknown. Here we show that an endogenously produced and secreted fungal oxylipin, 5,8-diHODE, induces fungal cellular differentiation, including lateral branching in pathogenic *Aspergillus fumigatus* and *Aspergillus flavus*, and appressorium formation in the rice blast pathogen *Magnaporthe grisea*. The *Aspergillus* branching response is specific to a subset of oxylipins and is signaled through G-protein coupled receptors. RNA-Seq profiling shows differential expression of many transcription factors in response to 5,8-diHODE. Screening of null mutants of 33 of those transcription factors identifies three transcriptional regulators that appear to mediate the *Aspergillus* branching response; one of the mutants is locked in a hypo-branching phenotype, while the other two mutants display a hyper-branching phenotype. Our work reveals an endogenous signal that triggers crucial developmental processes in filamentous fungi, and opens new avenues for research on the morphogenesis of filamentous fungi.

[1] Department of Medical Microbiology and Immunology, University of Wisconsin, Madison, Madison, WI, USA. [2] Department of Genetics, University of Wisconsin-Madison, Madison, WI, USA. [3] Department of Plant Pathology, University of Wisconsin—Madison, Madison, WI, USA. [4] Manchester Fungal Infection Group, Division of Infection, Immunity & Respiratory Medicine, University of Manchester, Manchester, UK. [5] Department of Biomedical Engineering, University of Wisconsin-Madison, Madison, WI, USA. [6] Department of Pharmaceutical Biosciences, Uppsala University, Uppsala, Sweden. [7] Department of Pathology and Laboratory Medicine, University of Wisconsin-Madison, Madison, WI, USA. [8] Department of Bacteriology, University of Wisconsin-Madison, Madison, WI, USA. ✉email: npkeller@wisc.edu

Colony development is characterized by complex hyphal networks in many fungi. While some yeasts and dimorphic fungi transit to or from hyphae only in selective conditions, filamentous fungi adopt a lifestyle with obligate, profuse, polarized hyphal growth primarily through apical extension and lateral branching[1,2]. Lateral branching mediates interhyphal communications through fusing with other hyphae, provides mechanical strength for substrate invasion, and allows for small molecular and nutrient exchange within the network[1,3]. Fungal mutants with decreased (hypo-) or increased (hyper-) branching often form morphologically aberrant colonies with compromised physiological integrity. Studies of *Aspergillus* mutants have shown that diverse cellular processes such as nuclear division[4], redox homeostasis[5], septum formation[6], endocytosis[7], cell wall integrity[8], and calcium[9] and Rho[10] signaling pathways can impact branching dynamics.

Although instructive, the complexity of these findings obfuscates a clear path to elucidating specific steps in this developmental program. We reasoned that the obligate requirement of branching in filamentous fungi, regardless of the environment, implicated the presence of endogenous metabolite(s) responsible for programmed hyphal branching. We hypothesized that a chemical class of endogenous oxygenated fatty acids, or oxylipins, might be involved in this program, as characterized *Aspergillus* oxylipin synthesis mutants are deviant in key developmental steps[11–14] (Fig. 1a) and oxylipins signal through G-protein-coupled receptors (GPCR)[15], a class of transmitters of fungal developmental signals[16]. Ppo proteins are fungal cyclooxygenase-like enzymes that are conserved across filamentous fungi, but absent in yeasts[17]. While the oxylipin products of these enzymes have been identified[18], no function has been assigned to these oxylipins in part due to difficulties in obtaining sufficient quantities for rigorous experimentation. Fungal oxylipins are not commercially available, but with advances in synthesis, analysis, and purification of these oxylipins[19,20], we obtained sufficient quantities of PpoA generated oxylipins to address our hypothesis that these metabolites act as developmental signals in filamentous fungi.

Here, we demonstrate that the secreted PpoA oxylipin 5,8-dihydroxyoctadecadienoic acid (5,8-diHODE) induces lateral hyphal branching in two pathogenic *Aspergillus* species, *A. fumigatus* and *A. flavus*. Screening of a panel of *A. flavus* GPCR mutants and *A. fumigatus* transcription factor mutants identified through 5,8-diHODE RNA-Seq profiling supports a model of an autocrine-like mechanism regulating hyphal branching in these species. Further, we find cross-genera recognition of fungal dihydroxyl oxylipins between *Aspergillus* and the rice blast pathogen *Magnaporthe grisea* that contains a PpoA-like enzyme[21]. However, rather than hyphal branching, the 5,8-diHODE signal induces the formation of appressoria in *M. grisea*, the structures required for fungal ingress of rice leaves. Together, our study has revealed a class of fungal signaling molecules and their profound impact on cellular differentiation processes and developmental steps critical for human and plant pathogens. More specifically, this work uncovers heretofore unknown signaling and regulatory molecules and pathways that regulate hyphal branching in pathogenic *Aspergillus* spp., allowing for an advance in the understanding of the largely unknown regulatory circuit of polarized hyphal growth.

## Results

**A. fumigatus Af293 produces a dihydroxyl oxylipin that inhibits sporulation.** Previous studies have identified the enzymatic activity and oxylipin products of PpoA using lyophilized fungal biomass[11]. We were able to assess the physiological levels of PpoA oxylipins in actively growing fungal cultures. We quantified the abundance of the two characterized PpoA linoleic acid-derived oxylipins, 8R-hydroxyoctadecadienoic acid (8R-HODE) and

5,8-dihydroxyoctadecadienoic acid (5,8-diHODE) in *A. fumigatus* Af293 wild-type (WT) and Δ*ppoA* cultures through ultra-high-performance liquid chromatography–tandem mass spectrometry (UHPLC-MS/MS; Supplementary Fig. 1). The Δ*ppoA* mutant did not produce 5,8-diHODE and produced barely detectable 8R-HODE (Fig. 1b). WT produced both metabolites where 5,8-diHODE but not 8R-HODE was secreted into the culture supernatant (Fig. 1c). Considering that 5,8-diHODE was absent in Δ*ppoA* and secreted in the WT strain, we hypothesized that the lack of this metabolite could contribute to the early sporulation phenotype of Δ*ppoA* (Supplementary Fig. 2a). We treated the WT and Δ*ppoA* with purified 5,8-diHODE or EtOH as the solvent control and found 5,8-diHODE restored the Δ*ppoA* sporulation level to that of WT in a concentration-dependent manner (Fig. 1d and Supplementary Fig. 2b).

**A. fumigatus Af293 oxylipin 5,8-diHODE induces hyphal branching.** A previous study found that PpoA co-localizes with the septin protein AspB at septa[22], the hyphal cross-walls in close proximity to lateral branches[23]. Thus, we asked if PpoA oxylipins had any effect on branching or septum formation. We developed a microfluidic-based assay that allowed for the visualization of growth dynamics of individual hyphae in microliters of culture volume over time (Supplementary Fig. 3). Incubation of *A. fumigatus* Af293 WT spores with 5,8-diHODE resulted in hyphae with excessive lateral branching coupled with stunted apical growth, leading to a hyperbranching morphology (Fig. 2a and Supplementary Movies 1 and 2). 5,8-diHODE-induced hyperbranching was dose-dependent with a significant increase in branching observed starting at 0.5 μg/mL (Fig. 2a, b), a physiologically relevant concentration (Fig. 1b). Staining with calcofluor white (CFW), a chitin-binding fluorescent dye, revealed shorter septal distances (Fig. 2c, d) and a higher level of chitin in 5,8-diHODE-treated hyphae (Fig. 2e), phenocopying other hyper-branched genetic mutants[9].

We found 5,8-diHODE-induced hyperbranching, regardless of whether it was introduced to spores (Fig. 2a, b) or hyphae (Fig. 2f). When hyphae were treated with 5,8-diHODE, induction of lateral branches was observed within 1.5 h of treatment (Fig. 2f). Since *A. fumigatus* strains are highly diverse in many physiological growth attributes[24], we also examined the effect of 5,8-diHODE in another commonly studied *A. fumigatus* strain, CEA10 (FGSC A1163), and found that it also branched in response to 5,8-diHODE at 5 μg/mL (Fig. 2g and Supplementary Movies 3 and 4).

As studies have shown that mitotic division, septation, and branching are temporally coordinated in *Aspergillus*[25,26], we evaluated if 5,8-diHODE affects nuclear division before or after the emergence of a lateral branch. Using time-lapse fluorescent microscopy, we tracked the dynamics of nuclear replication within a hyphal compartment and the lateral branch that newly emerged from it, from 1 h before until 4 h after lateral branch emergence (Supplementary Fig. 4a, b). When accounting for growth parameters, such as length and area of the branch, the number of nuclei per unit of area or length did not differ between the two groups (Supplementary Fig. 4c). These findings suggest that 5,8-diHODE did not lead to excessive branching through an altered nuclear division rate.

**Hyperbranching induction is specific to diol oxylipins and fungal species.** We were curious if the other PpoA linoleic acid-derived oxylipin, 8R-HODE, induced lateral branching and found it did not affect lateral branching at 5 μg/mL (Fig. 3a). Moreover, 8R-HODE reduced the efficacy of 5,8-diHODE on branching when treated with 5,8-diHODE concomitantly at 5 μg/mL,

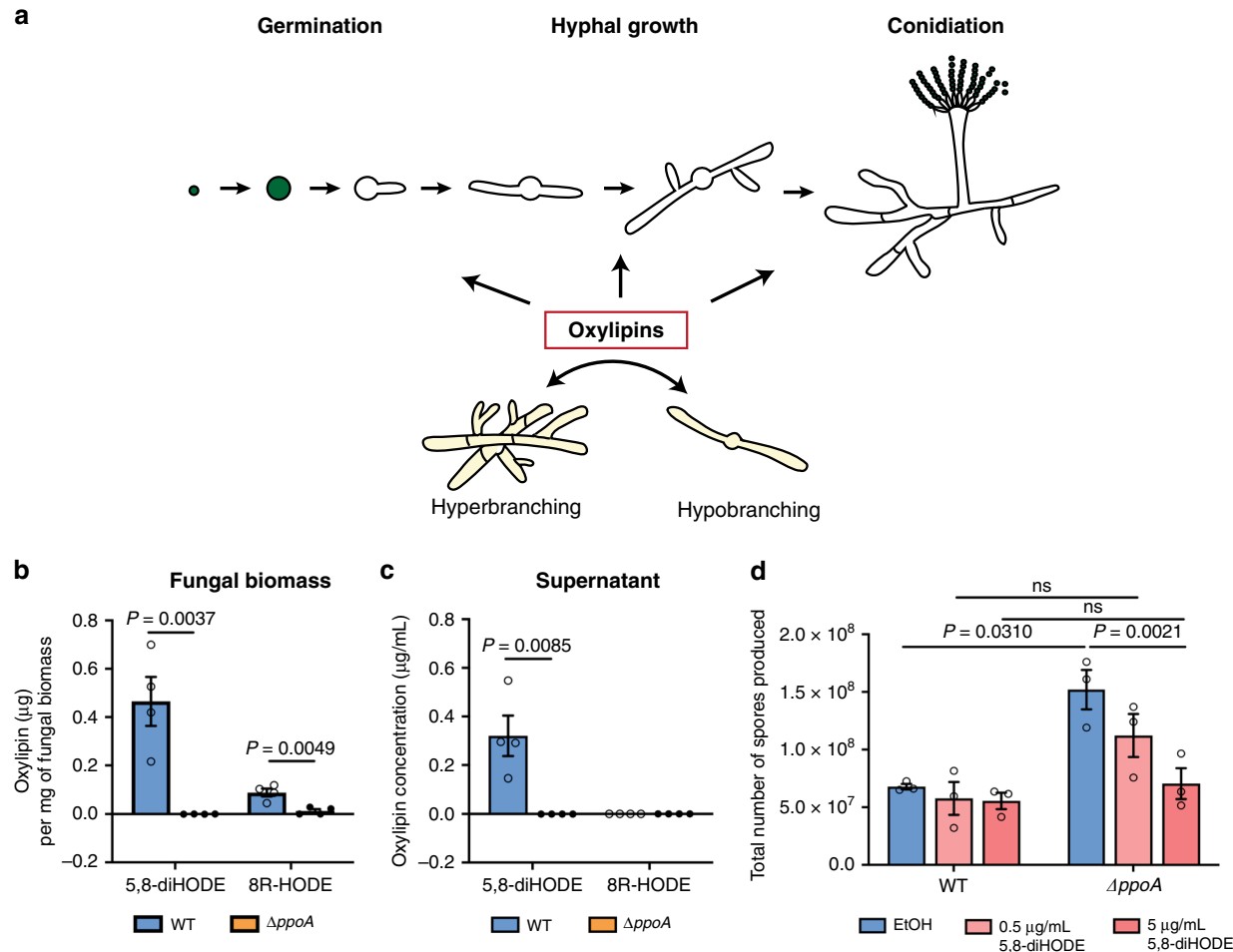

**Fig. 1 Production of PpoA oxylipins and complementation of ΔppoA sporulation phenotype by 5,8-diHODE. a** Developmental steps in *Aspergillus*. Asexual spores swell, germinate, form branched hyphae, and undergo conidiophore development. Different oxylipin signals are proposed to regulate germination, degree of branching, and development of spore-bearing conidiophores. **b** Quantification of PpoA oxylipins 5,8-diHODE and 8R-HODE in Af293 WT and ΔppoA fungal biomass, when cultures were grown at 25 °C for 120 h under constant shaking at 250 rpm (n = 4). Fungal biomass was separated from supernatant and extracted independently, followed with UHPLC-MS/MS analysis. **c** Quantification of PpoA oxylipins 5,8-diHODE and 8R-HODE in Af293 WT and ΔppoA culture supernatant obtained in the same experiment described in (**b**). **d** Sporulation quantification of Af293 WT and ΔppoA cultures (n = 3) treated in EtOH, 0.5 μg/mL and 5 μg/mL of 5,8-diHODE. Three independent cultures of Af293 WT and ΔppoA were grown in liquid shaking GMM at 25 °C for 65 h before the addition of 5,8-diHODE or EtOH and cultured up to 120 h before homogenization and spore quantification. A two-way ANOVA test followed by multiple comparison tests was performed to detect the difference between WT and ΔppoA in the three different treatment conditions. Multiple two-sided *t* tests were performed to compare the production of each oxylipin in WT and ΔppoA in (**a**, **b**), and two-way ANOVA was performed to test the effect of strain and treatment, followed by Holm–Šídák multiple comparison tests to compare two strains with the same treatment or two treatments in the same strain in (**c**). All values represent mean ± SEM. ns not significant (*P* > 0.05).

suggesting a possible competitive and antagonistic role of 8R-HODE against 5,8-diHODE activity. While we could not find any other report of an endogenous metabolite that induced branching in filamentous fungi, a former pioneering study demonstrated that plant natural products, specifically plant strigolactones, induced lateral branching in arbuscular mycorrhizae[27]. We tested the synthetic strigolactone analog GR24 and found no impact of GR24 on *Aspergillus* branching (Fig. 3b).

The results from 8R-HODE and GR24 suggested that *Aspergillus* may require structural specificity in oxylipins for recognition. To examine if hyperbranching is only induced by certain oxylipin chemical signatures, we tested a range of dihydroxyl and epoxidated oxylipins for hyphal branching induction in *A. fumigatus* Af293. We found two additional fungal dihydroxyl oxylipins, 5,8-dihydroxyoctadecenoic acid (5,8-diHOME, an oxylipin derived from oleic acid and produced by *Aspergillus* spp.[18]) and 7,8-dihydroxyoctadecadienoic acid (7,8-

diHODE, an oxylipin produced by the plant pathogen *Magnaporthe grisea* but not *Aspergillus* spp.[21]), induced hyperbranching behavior similar to that of 5,8-diHODE (Fig. 3c). However, other examined oxylipins had no impact on branching, suggesting of some required structural specificity, likely associated with the placement of hydroxyl groups, the number of double bonds, and/ or acyl chain length.

As many filamentous fungi undergo hyphal lateral branching, we were curious if a conserved endogenous signal, such as 5,8-diHODE, regulates this process in other fungi. We tested branching responses to 5 μg/mL 5,8-diHODE in several ascomycete species that either produce 5,8-diHODE or similar dihydroxyl oxylipins (*A. fumigatus, A. nidulans, A. flavus,* and *M. grisea*)[13,18,28] or contain a PpoA ortholog (*Penicillium expansum* XP_016594629, *Botrytis cinerea* XP_024548417). We found that only *A. fumigatus* and *A. flavus* showed increased branching in the presence of 5,8-diHODE (Fig. 4a). Intriguingly,

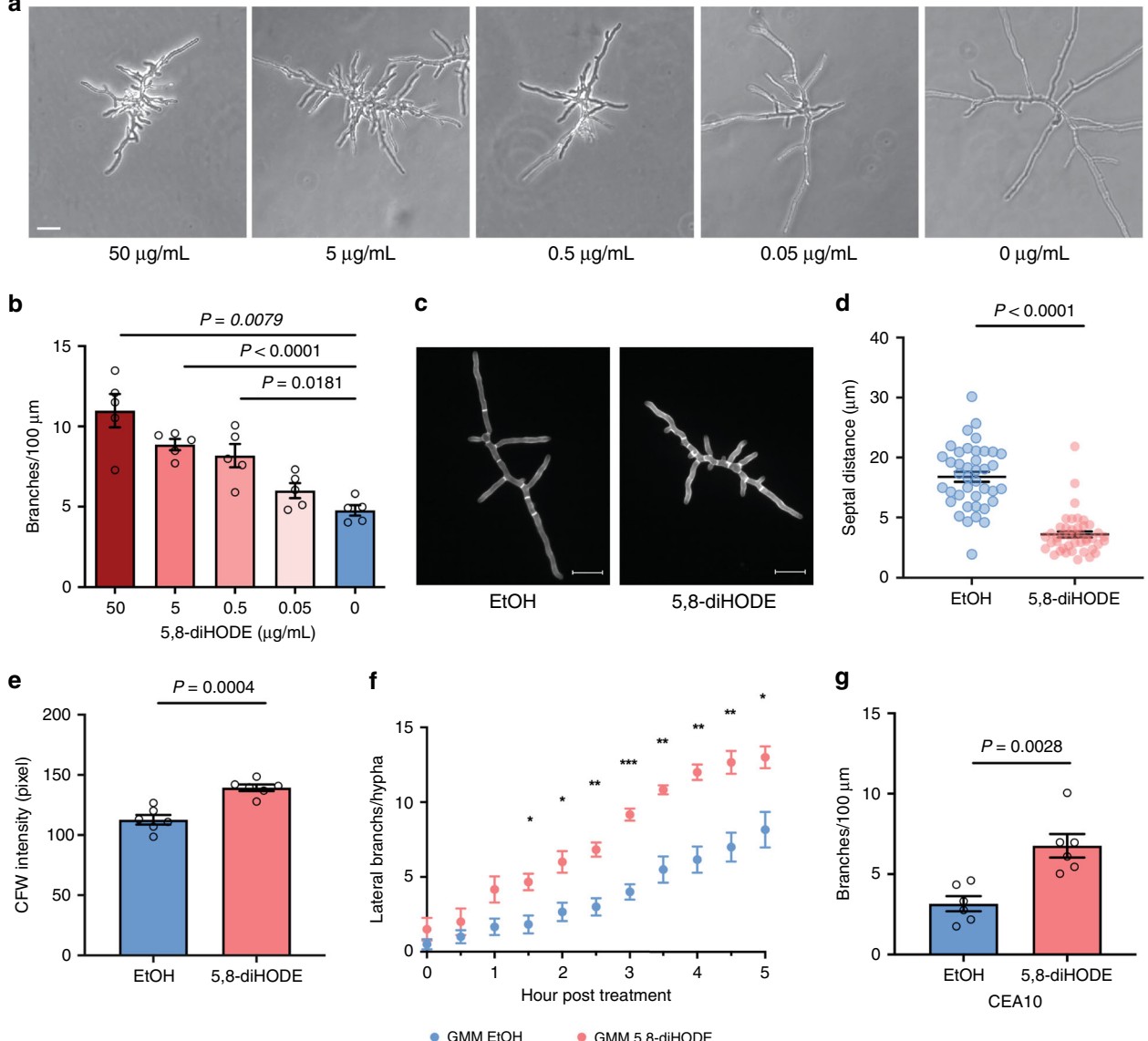

**Fig. 2 5,8-diHODE induces lateral branching and increased septa formation in *A. fumigatus* wild-type strains. a** Microscopic images of *A. fumigatus* Af293 WT grown in GMM containing either EtOH (0 µg/mL) or 5,8-diHODE at 0.05, 0.5, 5, and 50 µg/mL for 20 h in microfluidic wells. Images are representative of five microscopic images acquired for each treatment. Scale bar represents 20 µm. **b** A dose–response curve of hyphal branching at different concentrations of 5,8-diHODE ($n = 5$). The degree of branching is represented by the number of lateral branches/100 µm apical hyphal length at 20 h post inoculation. Af293 spores were incubated with GMM containing EtOH or 5,8-diHODE for 15 h to germinate, and five randomly selected young hyphae across four microfluidic wells were tracked with time-lapse microscopy every 15 min for 5 h. **c** Representative images of calcofluor white (CFW)-stained Af293 WT hyphae grown in GMM with EtOH or 5 µg/mL 5,8-diHODE for 14 h in a 24-well plate. Images are representative of six microscopic images acquired for each treatment. Scale bars represent 20 µm. **d** Distances between adjacent septa in EtOH ($n = 40$) and 5,8-diHODE-treated hyphae ($n = 43$) when six randomly selected hyphae from three wells were analyzed. $n$ represents the number of septal distances. **e** Quantification of CFW signal in the six selected hyphae analyzed in (**d**). **f** The number of emerging lateral branches per hypha when growing hyphae of the Af293 WT strain were exposed to 5 µg/mL 5,8-diHODE ($n = 6$). Hyphae were grown in GMM containing EtOH for 15 h and transferred to GMM either with EtOH or GMM with 5 µg/mL 5,8-diHODE, followed by time-lapse imaging of six hyphae across four microfluidic wells. **g** Branching response to 5 µg/mL 5,8-diHODE in another commonly used *A. fumigatus* clinical isolate, CEA10 ($n = 6$). Six randomly selected hyphae grown in four microfluidic wells were imaged every 15 min for 5 h. Brown–Forsythe and Welch ANOVA tests were performed, followed by Dunnett's T3 multiple comparison test between each treatment group and the no oxylipin control group in (**b**). Welch's two-sided $t$ tests were used for two group comparisons in (**d**, **e**, **g**). Multiple two-sided $t$ tests were performed to compare between the treatment groups in each time point in (**f**). All values represent mean ± SEM.

when exposed to 5,8-diHODE, *M. grisea* germlings differentiated predominantly into appressoria, infectious structures required for plant penetration[29] (Fig. 4b). Considering that *M. grisea* produces the branching-inducing oxylipin 7,8-diHODE identified in our screen (Fig. 3c)[21], we propose that 7,8-diHODE acts as an autocrine signal for appressorium formation in *M. grisea*.

Taken together, our data suggest that 5,8-diHODE and related dihydroxyl oxylipins can serve as signals for cellular differentiation processes that are manifested in different forms, including hyphal lateral branching and appressorium formation, and that these metabolites may act as paracrine signaling molecules between some fungal species.

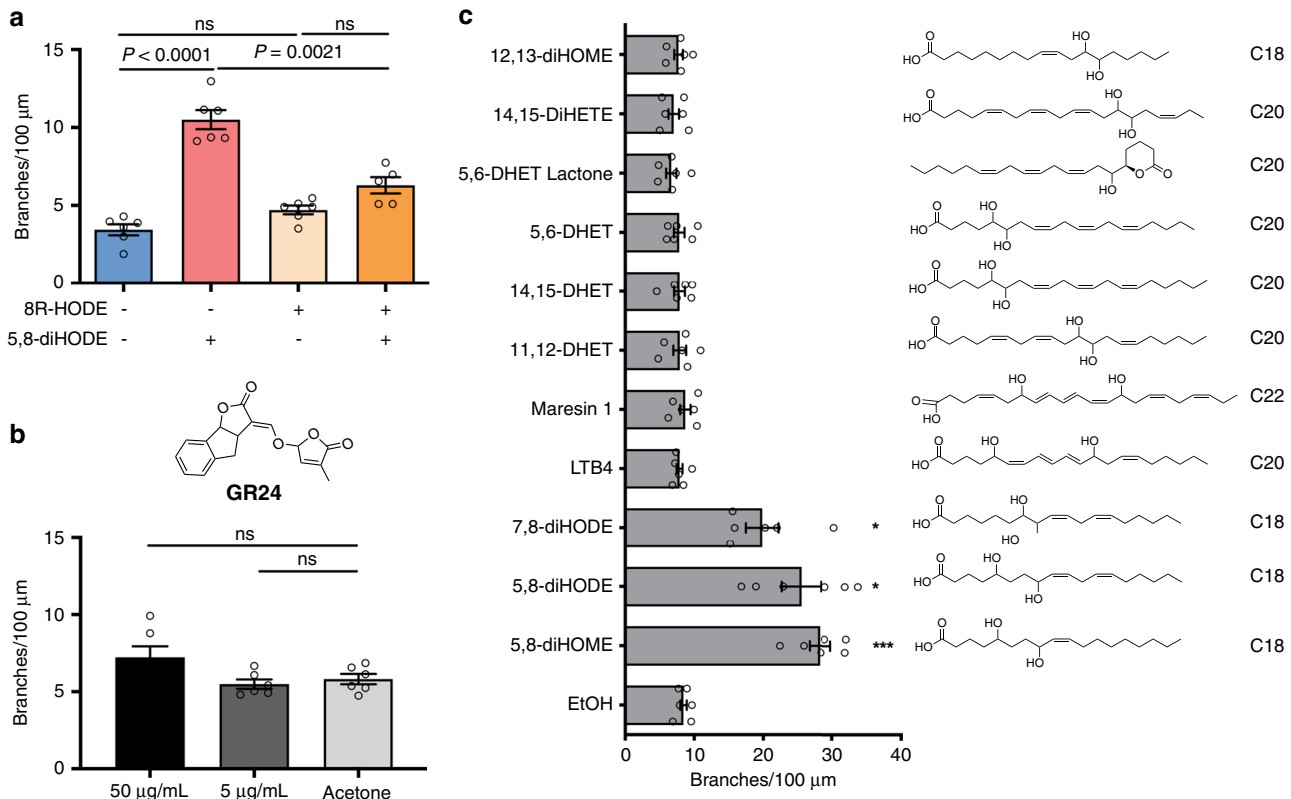

**Fig. 3 Hyperbranching is induced by specific diol oxylipins. a** Hyphal-branching analysis of Af293 WT when 8R-HODE (5 μg/mL) alone, 5,8-diHODE (5 μg/mL) alone, and 8R-HODE and 5,8-diHOE (both at 5 μg/mL) were treated to spores and grown for 20 h in microfluidic wells (n = 6 for no oxylipin, 8R-HODE alone, 5,8-diHODE, and n = 5 for 5,8-diHODE + 8R-HODE). **b** Branching analysis when GR24, a synthetic strigolactone, was treated to Af293 WT at 50 or 5 μg/mL compared to the acetone solvent control (n = 6). **c** Screen of a panel of dihydroxy oxylipins on Af293 WT for hyphal branching (n = 6). For all experiments, six hyphae were randomly selected across microfluidic wells and imaged every 15 min between 20 h and 25 h post inoculation. Compounds were either commercially purchased or purified in the laboratory and resuspended in EtOH. Brown–Forsythe and Welch ANOVA tests were performed, followed by Dunnett's T3 multiple comparison test in between each group in (**a**), and between the solvent control and each treatment group in (**b**, **c**). All values represent mean ± SEM. P values corresponding to asterisks from top to bottom in (**c**) are: 0.0372, 0.0146, 0.0001. ns not significant (P > 0.05).

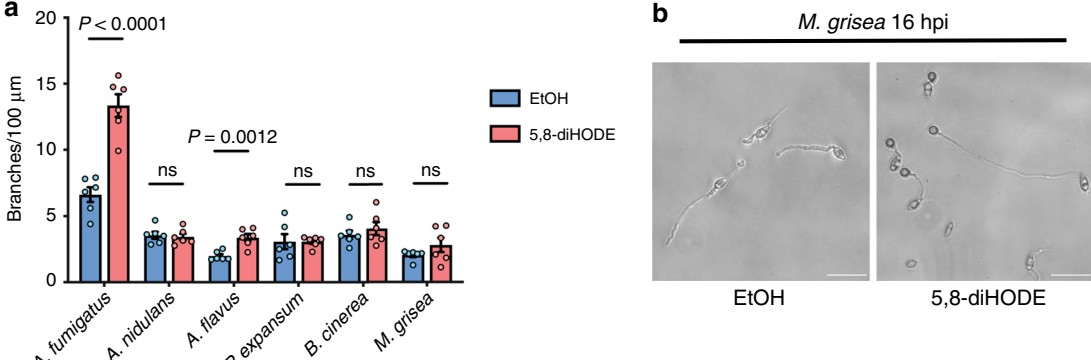

**Fig. 4 Branching response of several filamentous fungal species that contain a PpoA ortholog. a** Conidia of tested fungal species were inoculated in either GMM containing 1% EtOH or 5 μg/mL 5,8-diHODE and incubated in 96-well plates (n = 6). Six germinated hyphae were imaged via time-lapse microscopy for each treatment in each strain, and branching was assessed at specific time points when both apical and lateral growth were evident. All cultures were incubated at the optimal and permissive temperature specific to the species, and branching quantification was performed at a stage with pronounced apical growth and a significant degree of lateral branching. Multiple two-sided t tests were used to compare between treatment groups in each fungus. **b** Images of *M. grisea* cultures after exposure to either EtOH (1%) or 5 μg/mL 5,8-diHODE for 16 h. Images are representative of six microscopic images acquired for each condition. Scale bar represents 20 μm. All values represent mean ± SEM. ns not significant (P > 0.05). A *Aspergillus*, P *Penicillium*, B *Botrytis*, M *Magnaporthe*.

**5,8-diHODE signal is mediated through GPCRs.** Fungi possess a diverse class of GPCRs, transmembrane proteins that sense and convert environmental cues to intracellular signals that coordinate cellular responses[16,30]. GPCRs are primary receptors for mammalian oxylipins[31] and mediate the effects of exogenous plant oxylipins on *Aspergillus* spp.[15,32]. Since 5,8-diHODE induces excessive branching in both *A. fumigatus* and *A. flavus*, we utilized a previously constructed *A. flavus* GPCR mutant library[32] to ask if any GPCRs were involved in 5,8-diHODE-induced branching. Δ*gprC*, Δ*gprG*, and Δ*gprM* did not show increased branching when treated with 5 μg/mL 5,8-diHODE, while deletion of *gprG* led to increased branching at baseline (Supplementary Fig. 5). GprC and GprG in *A. flavus* have been found to be involved in sensing the plant oxylipin 13-hydroperoxyoctadecadienoic acid (13-HpODE)[32].

**Exposure of 5,8-diHODE leads to transcriptomic changes in *A. fumigatus*.** Identification of 5,8-diHODE as a branching signal offered an opportunity to interrogate the molecular pathways and

genetic network that govern hyphal branching in *A. fumigatus*. We performed a transcriptome profiling experiment to identify genes differentially expressed when *A. fumigatus* Af293 was treated with 5,8-diHODE (5 μg/mL) for 30 and 120 min. More than 4000 genes were differentially expressed at both time points (Fig. 5a, FDR < 0.05; Supplementary Data 1). In all, 30-min post exposure to 5,8-diHODE led to up- and downregulation of 361 and 94 genes, respectively ($|log_2FC| > 1$; Fig. 5a). In all, 120-min post exposure to 5,8-diHODE led to 450 upregulated genes and 119 downregulated genes ($|log_2FC| > 1$, Supplementary Fig. 6a). Gene Ontology analysis revealed a significant overrepresentation of annexin ($Ca^{2+}$-regulated phospholipid-binding and membrane-binding proteins) encoding genes upregulated at 30 min, secondary metabolite genes upregulated at 30 and 120 min, and genes with annotated transporter activities downregulated at 120 min (Supplementary Figs. 6 and 7, and Supplementary Data 2). Also, supporting the increase of CFW signal in 5,8-diHODE treatment (Fig. 2e), many chitin synthesis genes are

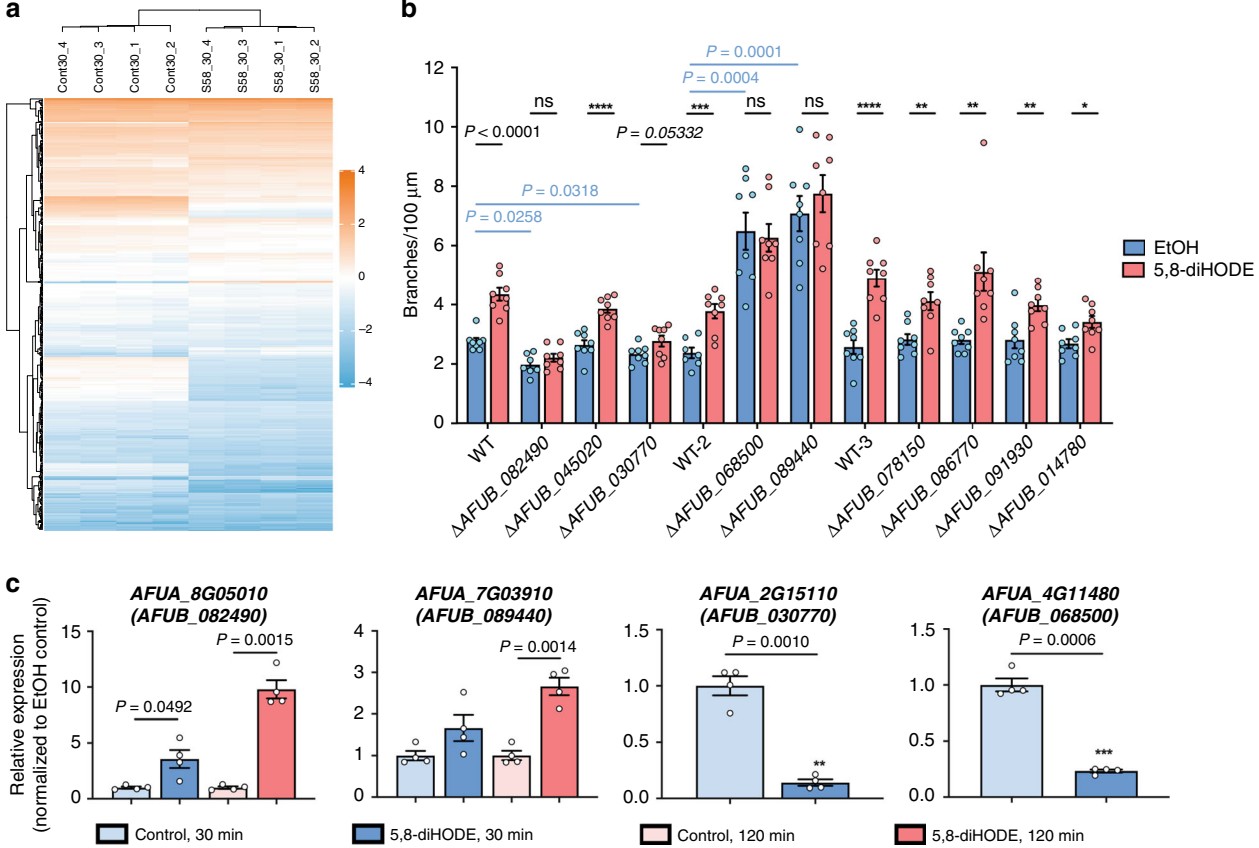

**Fig. 5 Global response to 5,8-diHODE and transcriptional regulators involved in 5,8-diHODE-induced branching. a** Hierarchical clustering analysis of differentially expressed genes (DEGs) with FDR < 0.05 and $|log_2FC|>1$ at 30 min post treatment from RNA-seq experimentation. The $log_2$FPKM (fragments per kilobase of transcript per million mapped reads) were used to construct the heatmap using the ComplexHeatmap package in RStudio. Cont_30: EtOH controls at 30 min; S58_30: 5,8-diHODE treated samples at 30 min. **b** Microfluidic-based screen of transcription factor (TF) mutants ($n = 7$ for Δ*AFUB_082490* and WT-2 in EtOH group, and $n = 8$ for the rest). Nine TF mutants from the first round of the screen were evaluated for branching in GMM containing EtOH or 5,8-diHODE (5 μg/mL) in four microfluidic wells in three separate batches. A1160 *pyrG*+ WT control was used as a positive control for each batch. For each treatment group in each strain, eight hyphae randomly selected from four microfluidic wells were imaged every 15 min for 5 h. *P* values corresponding to asterisks (*) from left to right are: <0.0001, 0.0005, <0.0001, 0.0026, 0.0038, 0.0041, 0.0105. **c** Expression of *AFUA_8G05010* and *AFUA_7G03910* at 30- and 120 min post treatment and *AFUA_2G15110* and *AFUA_4G11480* at 30 min post treatment ($n = 4$) through quantitative real-time PCR (qRT-PCR). The total RNA was extracted from four separate cultures that were either exposed to 0.005% EtOH or 5,8-diHODE (5 μg/mL), digested with DNase, and reverse-transcribed for SYBR-based real-time PCR. Expression of each gene in 5,8-diHODE-treated samples was normalized to its expression in the respective EtOH controls, which is set to 1. Multiple two-sided *t* tests were used to compare between treatments within each strain while Brown–Forsythe and Welch ANOVA tests were performed, followed by Dunnett's T3 multiple comparison test between strains within in EtOH group in (**b**). Welch's two-sided *t* tests were used to compare between treatment groups within the same time point in (**c**). All values represent mean ± SEM. ns not significant (*P* > 0.05).

significantly differentially regulated by 5,8-diHODE treatment in the RNA sequencing data (Supplementary Data 1).

**A genetic screen identifies transcription factors mediating branching response.** To quickly identify the regulatory pathways mediating lateral branching, we decided to focus on the 35 differentially expressed genes (DEGs) that encode putative transcription factors (TFs, $|\log_2 FC| > 1$; Supplementary Table 1). We identified 33 deletion mutants of the orthologs of these 35 genes in the *A. fumigatus* TFKO library (generated in the CEA10-derivative MFIG001 or A1160 $\Delta ku80$ $pyrG^+$)[33] and performed a screen of branching responses (no ortholog of *AFUA_3G02590* was found in CEA10, and the ortholog of *AFUA_6G07010* was not in the TFKO library). A first screen using 96-well plates identified nine deletion mutants statistically less responsive to 5,8-diHODE than WT as measured by hyphal branching (Supplementary Fig. 8a, b). To more accurately measure lateral branching, these nine mutants were subject to another round of screen using the microfluidic device with improved clarity compared to well plates, which identified three mutants that did not show increased branching and one mutant marginally responsive to 5,8-diHODE treatment at 5 μg/mL (Fig. 5b). We confirmed transcriptional changes of these four genes from the RNA-seq results with quantitative RT-PCR in Af293: *AFUA_8G05010* (*AFUB_082490* in CEA10) and *AFUA_7G03910* (*AFUB_089440* in CEA10) were significantly upregulated, and *AFUA_4G11480* (*AFUB_068500* in CEA10) and *AFUA_2G15110* (*AFUB_030770* in CEA10) were significantly downregulated (Fig. 5c and Supplementary Table 2).

$\Delta AFUB\_089440$ and $\Delta AFUB\_068500$ presented a hyperbranching phenotype (Fig. 5b). Both TFs have been partially characterized in other *Aspergillus* species. *nsdC* (*AFUA_7G03910/AFUB_089440*) encodes for a C2H2 TF ortholog of the *A. nidulans* and *A. flavus* NsdC that is required for suppressing asexual gene expression and promoting sexual stage formation in these species[34,35]. We observed the rapid development of conidiophores in this mutant, suggesting that *nsdC* suppresses *A. fumigatus* conidiation (Supplementary Movie 5). *AFUA_4G11480/AFUB_068500* encodes for a homolog of *A. nidulans* AslA, a C2H2 zinc finger TF that is required for potassium stress-induced hyperbranching and vacuolar biogenesis[36]. The deletion mutant of a third TF, *AFUA_8G05010/AFUB_082490*, which encodes for ZfpA, displayed a hypobranching phenotype (Fig. 5b). ZfpA is a C2H2 zinc finger TF of which the expression is induced by voriconazole[37] and high calcium[38]. The deletion of the fourth gene *AFUA_2G15110/AFUB_030770* resulted in a marginal response to 5,8-diHODE (Fig. 5b). This gene has not been characterized in any fungus to our knowledge.

**Transcription factor deletion mutants are aberrant in septation and cell wall integrity.** Since *A. fumigatus* WT hyphae exposed to 5,8-diHODE showed increased cell wall chitin deposition and decreased distance between septa (Fig. 2c–e), we asked if hyphae of the four TF mutants were aberrant in these aspects. Calcofluor staining intensity was slightly reduced in $\Delta AFUB\_082490/zfpA$ and significantly increased in $\Delta AFUB\_089440/nsdC$ compared to the WT in the EtOH control group (Fig. 6a, b). All strains showed significantly increased CFW signal when treated with 5,8-diHODE (Supplementary Fig. 9a). Septal distances were also altered in the mutants except for $\Delta AFUB\_030770$, compared to the WT in the EtOH control (Fig. 6c). The hypobranching mutant $\Delta AFUB\_082490/zfpA$ produced visually thin and elongated hyphae with septa further apart, while the hyperbranching strains $\Delta AFUB\_068500/aslA$ and $\Delta AFUB\_089440/nsdC$ showed reduced septal distances than WT (Fig. 6a, c). When treated with 5,8-diHODE compared to EtOH, hyphae of all strains but

$\Delta AFUB\_082490/zfpA$ showed altered septal distances (Supplementary Fig. 9b).

Lastly, considering the septation and chitin aberrancies noted above, we asked if any of these mutants were differentially sensitive to cell wall perturbation agents. We grew the TF deletion mutants in the presence of the cell wall stressor Congo Red and caspofungin, an antifungal and β-1,3-glucan synthase inhibitor that induces excessive branching and chitin biosynthesis in *A. fumigatus*[39,40]. $\Delta AFUB\_082490/zfpA$ showed increased sensitivity to both agents in comparison to WT (Fig. 6d).

**Overexpression of *AFUB_082490/zfpA* phenocopies 5,8-diHODE-treated wild-type.** To further confirm the role of specific TFs in 5,8-diHODE-induced hyperbranching, we overexpressed the two 5,8-diHODE upregulated TFs, *AFUB_082490/zfpA* and *AFUB_089440/nsdC*, in the CEA10 strain background (Supplementary Fig. 10a–c). We hypothesized that the overexpression strain would phenocopy the oxylipin-treated wild-type *A. fumigatus* if either of these TFs were critical for regulating lateral branching. We assessed multiple transformants of OE::*AFUB_082490/zfpA* and OE::*AFUB_089440/nsdC* for their branching phenotype and response to 5 μg/mL 5,8-diHODE in the 96-well format. All OE::*AFUB_082490/zfpA* mutants showed hyperbranching and were nonresponsive to 5,8-diHODE (Supplementary Fig. 11a, b). OE::*AFUB_089440/nsdC* mutants showed either comparable or slightly elevated branching compared to the WT without 5,8-diHODE, and marginally or significantly increased branching when treated with 5,8-diHODE (Supplementary Fig. 11a, b).

We further characterized one transformant of each of the overexpression strains (TNLR 28.1, OE::*AFUB_082490/zfpA*, and TNLR 29.6, OE::*AFUB_089440/nsdC*), through fluorescent microscopy. The data confirmed the hyperbranching phenotype of OE::*AFUB_082490/zfpA* and that OE::*AFUB_089440/nsdC* remained responsive to 5,8-diHODE (Fig. 7a, b). CFW signal was increased only in OE::*AFUB_082490/zfpA*, while septal distances were decreased in both overexpression strains compared to the WT EtOH control (Fig. 7c, d). 5,8-diHODE treatment did not affect OE::*AFUB_082490/zfpA* septal distance but resulted in further reduction of septal distance in OE::*AFUB_089440/nsdC* (Fig. 7e). CFW signal was further increased in 5,8-diHODE exposed hyphae of both mutants (Supplementary Fig. 11c). Taken all of these findings together, we conclude that ZfpA is a critical member of the 5,8-diHODE response cascade in regulating hyphal branching in *A. fumigatus* whereas NsdC, while impacting branching, may do so through other indirect pathways and may mediate other effect(s) of 5,8-diHODE such as suppression of sporulation (Fig. 1d)[34,35].

## Discussion

Here, we report that an endogenously produced and secreted hydroxyl oxylipin, 5,8-diHODE, directs cellular differentiation processes in filamentous fungi, including hyphal branching and appressorium formation. The *Aspergillus* branching response to oxylipins shows chemical specificity and is mediated through GPCRs. We used a combination of transcriptomic, genetic and cellular analyses that identified transcription factors promoting and suppressing oxylipin-induced hyphal branching and related cell wall processes, including hyphal septation, chitin deposition, and response to cell wall perturbation agents. This resulted in the identification of a C2H2 zinc finger protein, ZfpA, that regulates lateral branching in *A. fumigatus*. Taken together, our work supports an oxylipin autocrine/paracrine model where specific oxylipin ligands and select GPCRs direct fungal differentiation decisions (Fig. 8).

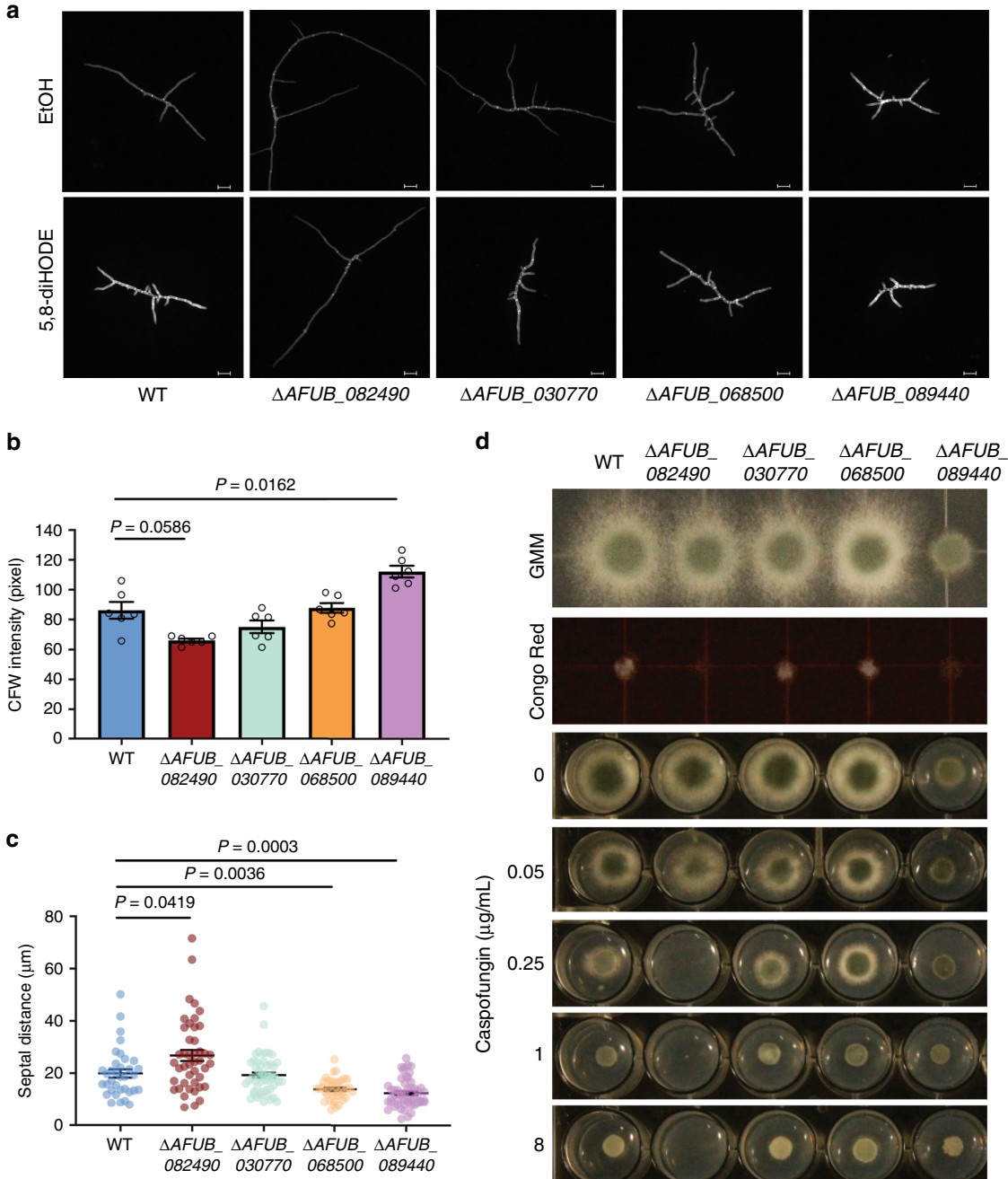

**Fig. 6 5,8-diHODE-responsive transcription factor deletants display altered hyphal septation and cell wall integrity. a** Representative DAPI images of *A. fumigatus* A1160 *pyrG*⁺ WT control and deletion mutants Δ*AFUB_082490*, Δ*AFUB_030770*, Δ*AFUB_068500*, and Δ*AFUB_089440* grown in GMM containing 0.1% EtOH or 5 μg/mL 5,8-diHODE. Fourteen-hour-old hyphae grown on a coverslip in a 24-well plate were stained with calcofluor white (CFW), washed in ddH₂O, and mounted to slides. Six DAPI images were acquired from three wells for each condition in each strain (*n* = 6). Images are representative of six microscopic images acquired for each condition. Scale bars represent 20 μm. **b** Quantification of CFW signals of hyphae (*n* = 6) grown in GMM + 0.1% EtOH from the six DAPI images acquired in (**a**). Hypha in each DAPI image was selected through the thresholding function in FIJI, and the mean fluorescence intensity of each threshold area was measured. **c** Quantification of distances between adjacent septa in the six hyphae of each strain grown in GMM + 0.1% EtOH and imaged in the experiment performed in (**a**). Sample sizes representing the number of septal distances counted for each strain are: *n* = 35 for WT, *n* = 44 for Δ*AFUB_082490*, *n* = 51 for Δ*AFUB_030770*, *n* = 40 for Δ*AFUB_068500*, *n* = 52 for Δ*AFUB_089440*. **d** Sensitivity of A1160 *pyrG*⁺ WT control and transcription factor deletion mutants to cell wall perturbation agents Congo Red (25 μg/mL) and caspofungin at indicated concentrations. GMM was used as control. In all, 2000 spores were inoculated and grown for 2 days before the visual examination. Brown–Forsythe and Welch ANOVA tests were performed, followed by Dunnett's T3 multiple comparison test between mutant strains and the WT control in (**b**, **c**). All values represent mean ± SEM.

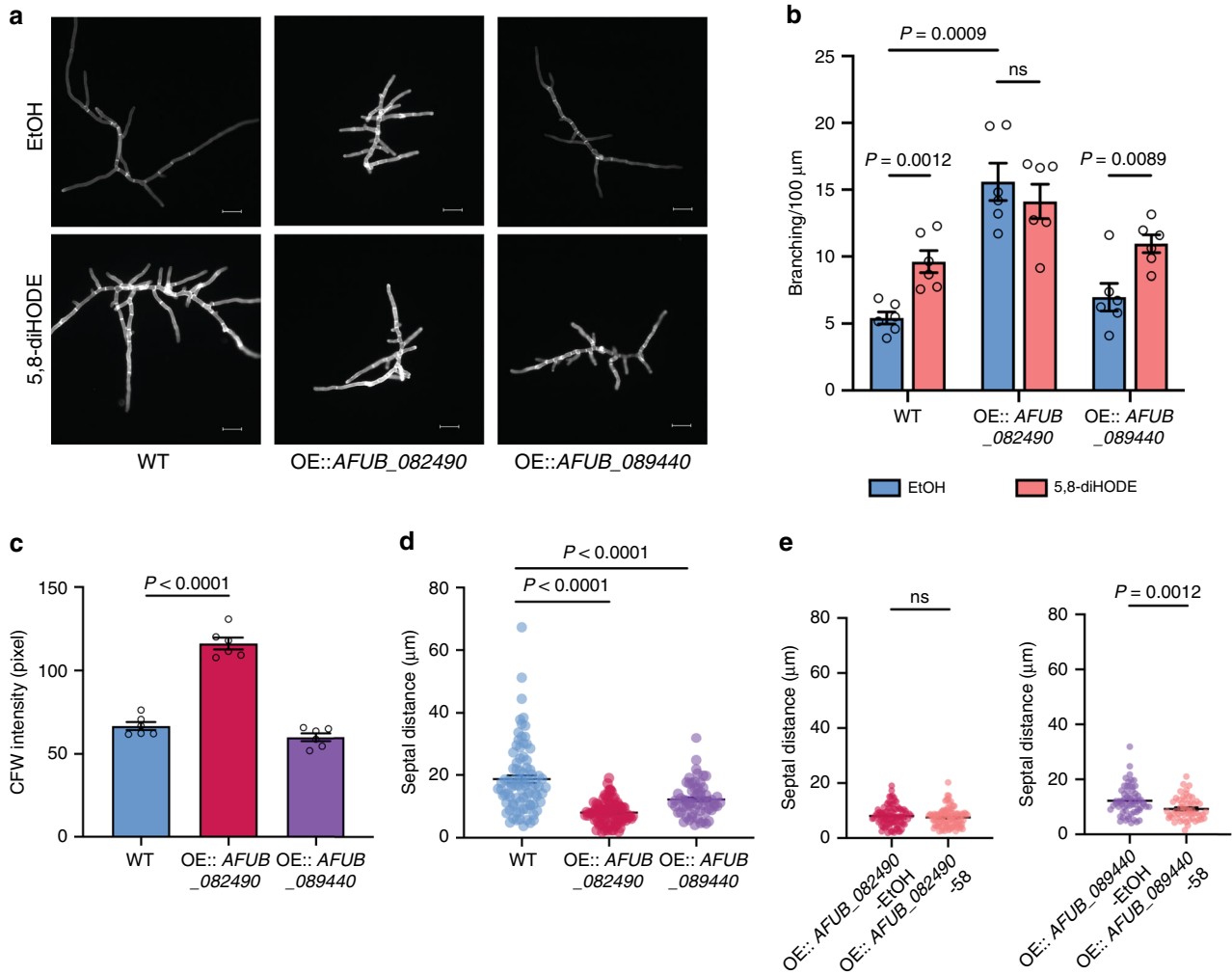

**Fig. 7 Overexpression of *AFUB_082490* and *AFUB_089440* reveals major involvement in 5,8-diHODE-induced branching. a** Representative DAPI images of A1160 *pyrG+* WT, OE::*AFUB_082490* and OE::*AFUB_089440* grown in GMM containing 0.1% EtOH or 5 μg/mL of 5,8-diHODE on coverslips in 24-well plates. Sixteen-hour-old hyphae were stained with calcofluor white (CFW), washed in ddH2O, and mounted to slides for fluorescent microscopy. Six DAPI images were acquired from three wells for each condition in each strain ($n = 6$). Images are representative of six microscopic images acquired for each condition. Scale bars represent 20 μm. **b** Branching quantification of OE::*AFUB_082490* and OE::*AFUB_089440* and the A1160 *pyrG+* WT control ($n = 6$) in the experiment conducted in (**a**). Six hyphae from the six DAPI images were analyzed for branching for each condition in each strain. **c** Quantification of CFW signals ($n = 6$) of hyphae grown in GMM + 0.1% EtOH from images acquired in (**a**). Six hyphae in the six DAPI images were analyzed, where hyphae in each DAPI image were selected through the thresholding function in FIJI, and the mean fluorescence intensity of each threshold area was measured. **d** Quantification of distances between adjacent septa in hyphae grown in GMM + 0.1% EtOH from experiments conducted and imaged acquired in (**a**). Sample sizes representing the number of septal distances counted for each strain are: $n = 85$ for WT, $n = 90$ for OE::*AFUB_082490*, and $n = 61$ for OE::*AFUB_089440*. **e** Quantification of septal distances in OE::*AFUB_082490* and OE::*AFUB_089440* hyphae grown in GMM + 0.1% EtOH (OE-EtOH) and GMM + 5 μg/mL 5,8-diHODE (OE-58) in the same experiment described in (**a**). Sample sizes representing the number of septal distances counted for each strain are: $n = 90$ for OE::*AFUB_082490*-EtOH, $n = 92$ for OE::*AFUB_082490*-58, $n = 61$ for OE::*AFUB_089440*-EtOH, and $n = 53$ for OE::*AFUB_089440*-58. A1160 *pyrG+* WT was used as control for all experiments. Multiple two-sided *t* tests were used to compare between treatments in each strain in (**b**) and Brown–Forsythe and Welch ANOVA tests followed by Dunnett's T3 multiple comparison test were used to compare across strains within each treatment group in (**b**–**d**). Welch two-sided *t* tests were used to compare between treatments within each strain in (**e**). All values represent mean ± SEM. ns not significant ($P > 0.05$).

We identified 5,8-diHODE-directed developmental responses in three filamentous fungi in our experimental conditions. w?>Although this may suggest that this response is limited to specific fungi, possibly in a niche-specific manner (e.g., invasive growth of pathogenic *Aspergillus* spp. and appressoria requirement for *M. grisea* infections), we hypothesize that oxylipin-directed fungal differentiation may be widely present in nature. It is possible that many fungi may respond to different combinations of specific oxylipins, displaying different developmental responses, depending on the ecological context. Consistent with this hypothesis, PpoA and similar oxygenases are conserved in filamentous fungi, and their loss or overexpression impacts fungal development, even in species such as *A. nidulans* that did not display a branching response to 5,8-diHODE (Fig. 4)[11,13,14]. Furthermore, fungal recognition of plant and animal oxylipins results in morphological or chemotactic responses in the fungus[15,17,41]. Moreover, our finding that 8R-HODE inhibits 5,8-diHODE branching (Fig. 3a), and that *Aspergillus* GPCRs are important in 5,8-diHODE (Supplementary Fig. 5) and plant oxylipin recognition[15,32], is reminiscent of the

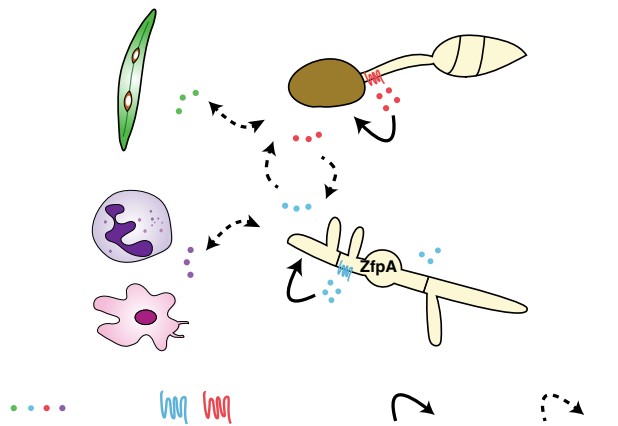

Oxylipins G-protein-coupled receptors Autocrine action Paracrine action

**Fig. 8 Model of autocrine and paracrine activities of fungal oxylipins.**
Autocrine activities of endogenous oxylipins, such as 5,8-diHODE produced in *A. fumigatus* and 7,8-diHODE produced in *M. grisea*, direct hyphal branching and appressorium formation in the respective fungal species. These oxylipins regulate cellular differentiation likely through G-protein-coupled receptors (GPCRs) and in *A. fumigatus*, through the transcription factor ZfpA. Secreted oxylipins can exert paracrine effects on neighboring organisms, including other fungi occupying a common niche and plant and animal host immune cells during fungal infection.

ligand-receptor-mediated signaling well characterized in mammalian eicosanoid/GPCR biology where different eicosanoids mediate inflammatory responses through GPCRs[42]. Thus, we hypothesize that oxylipins may act as signals in intra- and inter-Kingdom interactions between fungi and other organisms, possibly in a bi-directional manner (Fig. 8).

In *A. fumigatus*, PpoA is highly expressed and binds to the septum-localizing septin AspB in the presence of the antifungal caspofungin[22,43], suggesting of a localized production of 5,8-diHODE. It could be subsequently involved in antifungal responses involving an evasive tactic of lateral branching, which was also reported during *A. fumigatus* encounters with neutrophils[44]. Sensing of 5,8-diHODE by specific GPCRs would culminate in ZfpA activation and lateral branch formation (Fig. 8). The involvement of ZfpA in this model is supported by our finding that *zfpA* deletion mutant is several folds more sensitive to caspofungin (Fig. 6d), and that *zfpA* is highly induced by the fungicide voriconazole[37] and by high calcium[38]. Calcium is a key intracellular polar growth signal in *A. fumigatus*[45] and other filamentous fungi[46]; thus we hypothesize that calcium dynamics may be linked to the oxylipin/ZfpA transduction pathway uncovered in this study. We note that the expression of many calcium homeostasis genes, such as annexin-encoding genes, was affected by 5,8-diHODE treatment (Supplementary Fig. 7a, Supplementary Tables 1 and 2). We also recognize that hyphal branching is a highly complex process regulated and influenced by a series of internal and external factors, involving multiple molecular events at transcriptional, translational, and post-translational levels that are unlikely to be solely captured by our model (Fig. 8). We posit, however, that the identification of a fungal communication language that signals through oxylipin-responsive transcriptional regulators and GPCRs provides testable opportunities to identify processes involved in hyphal growth patterns of pathogenic Aspergilli. In future studies, adoption of more advanced microscopic tracking of hyphal growth and branching[47], coupled with localization of PpoA, GPCR, and ZfpA, would allow to better define their involvement in the spatiotemporal control of hyphal branching, as illustrated in other studies[48].

In conclusion, finding an endogenous developmental signal advances our understanding of the fungal morphogenesis. Although this work was focused on the oxylipin-induced hyphal branching response in *Aspergillus*, we predict that an analogous oxylipin-mediated GPCR cascade leading to appressorium formation may exist in the plant pathogen *M. grisea*[49]. Our work suggests that these metabolites may represent a common language regulating aspects of differential growth across a broad range of fungi. Filamentous fungi include human, plant, and animal pathogens, industrial workhorses, mycorrhizae, and saprophytic organic-degrading organisms. The identification of endogenous fungal signals paves the path to not only address gaps in the general understanding of cellular processes mediating multicellular growth in filamentous fungi but also provide strategies to study host tissue entry and dissemination by pathogens and symbionts.

## Methods

**Fungal strains and culture conditions.** *Aspergillus fumigatus* Af293 and CEA10 wild-type strains are ATCC available and common laboratory clinical isolates used in other studies[12,50]. Af293 $\Delta ppoA$ were used as in Dagenais et al.[12]. *A. fumigatus* strains were grown on glucose minimal medium (GMM) plates, conidia were collected in sterile water containing 0.01% Tween 80 and maintained as glycerol stocks at −80 °C. Other fungi assessed in the hyphal-branching assay were activated on media and conditions appropriate for the species before conidia collection following the same procedure. *Aspergillus nidulans* FGSC A4 and *Aspergillus flavus* NRRL 3357 (ATCC available) were cultured on GMM at 37 °C for 3 days and 29 °C for 4 days, respectively. *Penicillium expansum* P21 and *Botrytis cinerea* B05 were cultured on GMM at 25 °C for 5 days. *Magnaporthe grisea* Guy-11 was cultured on potato dextrose agar (PDA) at room temperature for 7 days. *A. flavus* GPCR deletion mutants and the isogenic wild-type control strain were used as published in Affeldt et al.[32]. All strains are available from Keller lab.

**Chemicals.** Purified 8R-HODE, 5,8-diHODE, 5,8-diHOME, and 7,8-diHODE were prepared as below[19]. To quantify production, a small aliquot was mixed with an internal standard (2 µg of 12-oxophytodienoic acid) and analyzed by reverse-phase high-performance liquid chromatography–tandem mass spectrometry (RP-HPLC-MS/MS); the intensities of the $[M-H]^-$ anions of the metabolites and the internal standard were used for quantification. A larger quantity of 5,8-diHODE was kindly donated by Dr. Oh from Konkuk University, South Korea[20]. All other oxylipins used in this study were commercially available and purchased from Cayman Chemical (Ann Arbor, Michigan). These include 12,13-dihydroxy-9Z-octadecenoic acid (12,13-diHOME), 14,15-dihydroxy-5Z,8Z,11Z,17Z-eicosatetraenoic acid (14,15-diHETE), 5,6-Dihydroxy-8Z,11Z,14Z-eicosatrienoic acid 1,5-lactone (5,6-DHET lactone), 5,6-dihydroxy-8Z,11Z,14Z-eicosatrienoic acid (5,6-DHET), 14,15-dihydroxy-5Z,8Z,11Z-eicosatrienoic acid (14,15-DHET), 11,12-dihydroxy-5Z,8Z,11Z-eicosatrienoic acid (11,12-DHET), maresin 1, and leukotriene B4 (LTB4). GR24 was a generous gift from Dr. Christopher McErlean at the University of Sydney.

**Oxylipin extraction.** WT and $\Delta ppoA$ spores were inoculated at $10^7$ spores/mL in liquid GMM and grown at 25 °C for 5 days, with constant shaking at 250 rpm. Supernatant and fungal biomass were separated via centrifugation at 4000 rpm for 15 min. In total, 25 mL of supernatant were directly used for extraction, and tissues were washed twice with ddH$_2$O before lyophilization. Total oxylipins were extracted using mixed organic solvents of ethyl acetate:methanol:dichloromethane (8:1:1) overnight[51]. The organic phase was separated and collected using a separatory funnel and evaporated to dryness using a Buchi Rotovap R-210. To extract from fungal biomass, lyophilized tissue was weighed, and ~20 mg of lyophilized fungal tissues were resuspended in ddH$_2$O with 0.1% formic acid, homogenized, and extracted. Extracts were resuspended in methanol and stored at −20 °C before analysis.

**UHPLC-MS/MS analysis.** Production of 8R-HODE and 5,8-diHODE was measured using ultra-high-performance liquid chromatography–tandem mass spectrometry (UHPLC-MS/MS). Samples were separated through the Zorbax Eclipse XDB-C18 column (2.1 × 150 mm, 1.8-µm particle size) at a flow rate of 0.2 mL/min on a Thermo Scientific-Vanquish UHPLC system connected to a Thermo Scientific Q Exactive Orbitrap mass spectrometer in negative mode. A 20-min gradient was employed using LC-MS grade water with 0.05% formic acid (solvent A) and LC-MS grade acetonitrile with 0.05% formic acid (solvent B) using the following gradient: 0 min, 55% solvent B; 2 min, 55% solvent B; 18 min, 98% solvent B; 20 min, 98% solvent B; 25 min, 55% solvent[52]. Full MS spectra were acquired at 70,000 resolution for the *m/z* range between 150 and 2000 for all samples. Following each full MS scan, the top five most intense ions were selected for a dependent MS² scan. MS² was conducted using higher-energy collisional dissociation with normalized collisional energy of 30.

Purified 8R-HODE and 5,8-diHODE of known serial concentrations were analyzed in the same analytical conditions to determine the limit of detection and construct a standard curve. At specific retention time intervals, the intensity of the characteristic daughter ions in the $MS^2$ scan for 8R-HODE ($m/z = 157.0859$, r.t. = 8–9 min) and 5,8-diHODE ($m/z = 173.0810$, r.t. = 3–4 min) were used to represent the absolute abundance of each oxylipin in biological samples[19]. All samples and standards were analyzed using the Thermo Xcalibur Qual Browser (Version 3.1.66.10). Each strain analyzed was assessed in a biological quadruplicate. The amount of 8R-HODE and 5,8-diHODE was normalized to supernatant volume dry biomass obtained from each culture.

**Sporulation assay**. Spores of ΔppoA and WT were inoculated and grown at the same conditions as in the oxylipin quantification assay, and asexual spore quantity was assessed at 120 h post inoculation after culture homogenization. To assess the effect of 5,8-diHODE on sporulation, 5,8-diHODE was added to the growing cultures to a final concentration of 0.5 μg/mL and 5 μg/mL, and EtOH was added to achieve a final concentration of 0.02% as the solvent control at 65 h post inoculation. All cultures were homogenized, and spores were enumerated using a hemocytometer after 120 h post inoculation.

**Microfluidic platform fabrication**. A microfluidic O-Channel platform was designed to allow for microscopic observations using minimal culture volume (Supplementary Fig. 3). The microfluidic arrays were fabricated following Berthier et al.[53] with the following modifications. The 1:10 mixture of PDMS and curing agent (Sylgaard 184, Dow Corning, USA) was poured onto the O-channel and fabricated at 60 °C for 6 h. During the experimental setup, the PDMS layer was peeled off and transferred inside a polystyrene Omnitray dish (NUNC, USA), plasma-treated for 2 min using a Unitronics device (Plasma Etch, Carson City NV), and UV-sterilized for 15 min before sample loading.

**Hyphal-branching assessment**. In total, 2500 spores with designated treatment in 5 μL GMM was inoculated into each O-channel well, and wet Kim wipes were placed against two sides of the Omnitray dish to prevent samples from drying out. Samples were incubated at 37 °C for 15 h in a heated microscope enclosure (OKO Labs, Burlingame, CA) developed for a Nikon Eclipse Ti inverted microscope. At least six germinating spores were randomly selected and monitored using a Nikon Plan Fluor 20x Ph1 DLL objective, and phase-contrast images were captured every 15 min for 5 h using the Nikon NIS Elements AR software package (Version 4.13). The length of the leading hypha and the number of primary lateral branches per leading hypha were quantified in the NIS Elements AR software package. Initial screen of A1160 TF mutants was performed in 96-well plates with 1000 spores per well, treated with either 1% EtOH or 5 μg/mL 5,8-diHODE. Eight hyphae were analyzed in randomly selected fields among the three replicate wells. Hyphal branching of other fungi was assessed in a 96-well plate when cultured at different temperatures and quantified at specific hour post incubation (hpi): 37 °C for A. nidulans at 16 hpi, 29 °C for A. flavus at 28 hpi, and 25 °C for P. expansum at 32 hpi, B. cinerea at 24 hpi, and M. grisea at 36 hpi. Image acquisition and data analysis were performed in the same manner as when assessing branching in A. fumigatus.

**Nuclear replication analysis**. GFP-histone-tagged A. fumigatus Af293 strain TJMP 131.5 were inoculated at $8 \times 10^3$ spores per 100 μL and grown in GMM at 37 °C for 7 h in a 96-well plate. To ensure synchronous cell division across strains, 20 mM hydroxyurea was added and incubated for 2 h at 37 °C. The cultures were then washed 3× with GMM. Finally, 5,8-diHODE was added to a final concentration of 5 μg/mL or ethanol at 1% final concentration in GMM. Upon settlement for 1.5 h, time-lapse imaging in both Phase and GFP channels was performed every 20 min for the next 15 h, using a Nikon Plan Fluor ×20 objective. To analyze nuclei replication during lateral branching, a new lateral branch was selected based on two criteria: the cell compartment where the branch emergence can be clearly identified, with the two adjacent septa visible; the GFP-labeled nuclei within the cell compartment and the lateral branch can be clearly tracked from 1 h before until 4 h after branch emergence. Six samples from either EtOH or 5,8-diHODE-treated group were selected and analyzed for the 5 h period. The length and area of the total region consisting of the lateral branch and the apical cell compartment were quantified at the time of branch emergence (T) and 4 h post this time point in the Nikon NIS Elements AR software package (Version 4.13).

**Septal distance and CFW signal quantification**. In total, 2500 spores were inoculated in 24-well plates with coverslips mounted to the bottom and cultured at 37 °C with or without 5,8-diHODE (5 μg/mL). Depending on the experiment, 14–16-h-old hyphae were stained with 0.1 mg/mL CFW, according to Juvvadi et al.[9]. Six hyphae were randomly selected for DAPI fluorescent imaging using the Nikon Plan Fluor ×20 Ph1 DLL objective. Distances between two adjacent septa were quantified in the Nikon NIS Elements AR software package (Version 4.13). DAPI images containing a single hypha were analyzed in FIJI (Version 2.0.0-rc-69/1.52p) for CFW fluorescence intensity quantification. All images were processed using the same Macro script that executes these functions in a sequential order: split channel, auto-threshold "RenyiEntropy dark" for areas with hyphae in the blue channel, and measurement of the mean intensity of the thresholding area.

**RNA extraction and quantitative RT-PCR**. A. fumigatus Af293 WT spores were inoculated at $10^6$ spores/mL, grown overnight in liquid GMM at 37 °C and 250 rpm, and treated with either 5,8-diHODE (5 μg/mL) or EtOH (0.005%) for 30 min and 120 min. Four biological replicates were included for each condition. At 30 min and 120 min post treatment, total fungal biomass was harvested, flash-frozen in liquid nitrogen, and lyophilized. The total RNA was extracted using QIAzol Lysis Reagent (Qiagen) according to the manufacturer's instructions with additional phenol:chloroform:isoamylalcohol (24:1:1) extraction step before RNA precipitation. For quantitative RT-PCR, RNA was digested with DNase I (New England Biolabs) and reverse-transcribed using iScript cDNA synthesis kit (Biorad). Quantitative PCR was performed using iQ SYBR Green Supermix (Bio-Rad) following the manufacture's cycle conditions in the CFX Connect Real-Time System machine (Bio-Rad), with 12.5 ng input cDNA and primers listed in Supplementary Table 4. The expression level of each gene was normalized to act1 expression level of the respective sample using the $2^{-\Delta\Delta C_T}$ (Livak) method[54].

**RNA sequencing and analysis**. To prepare for RNA sequencing, total RNAs were further cleaned up using RNeasy Mini Kit with on-column DNase digestion (Qiagen). RNA sequencing, library preparation, and differential expression analysis were performed by Novogene, Inc. RNA integrity was tested via nanodrop, gel electrophoresis, and in the Agilent 2100 bioanalyzer. Libraries were prepared using the TruSeq library preparation protocol with poly-A mRNA enrichment and pair-end sequencing (150 bp) in the Illumina Hiseq2500 system. Adapter and low-quality reads were removed, and clean reads were mapped to the annotated genome of A. fumigatus strain Af293 obtained from FungiDB (release 40) using TopHat (v2.0.12). The read-count table was processed through DESeq2 (v1.10.1) in R to identify differentially regulated genes between EtOH- and 5,8-diHODE-treated samples at both time points. HTSeq (v0.6.1) was used to calculate fragments per kilobase of transcript per millions mapped reads (FPKM), normalized gene expression. Heatmaps were drawn in RStudio (v1.1.463) using the packages zFPKM (v1.8.0) for log transformation and ComplexHeatmap (v2.2.0) for hierarchical clustering analysis and graphing, for DEGs with Benjamini–Hochberg false discovery rate (FDR) < 0.05, |log₂ fold change($\log_2$FC)|>1, and $\log_2$(FPKM) > −3. Gene Ontology of DEGs (FDR < 0.05 and |$\log_2$FC|>1) was analyzed for enrichment in FungiDB[55] (https://fungidb.org/fungidb/) and visualized as scatter plots in REVIGO[56].

**Transcription factor deletion and overexpression mutants**. Null mutants were generated in the A. fumigatus strain MFIG001 by directed replacement of the TF of interest with a hygromycin (hph) selectable marker as described in Zhao et al.[57] and validated using the processes outlined in Furukawa et al.[33]. Overexpression mutants of the transcription factor encoding the genes AFUB_082490 and AFUB_089440 were created following previously published protocols for over-expression cassette construction[58], protoplast generation and transformation[59]. The 5′ and 3′ flanking regions of the cassette were amplified from A. fumigatus CEA10 genome and fused with the fragment consisting of A. parasiticus pyrG and A. nidulans gpdA promoter amplified from pJMP9.1[60]. The product was used to transform the uridine/uracil auxotrophic strain CEA17 Δku80 ΔpyrG[61]. Resultant transformants were screened by PCR using primers listed in Supplementary Table 4.

**Cell wall perturbation sensitivity**. A. fumigatus A1160 pyrG⁺ and the transcription factor deletion mutants were grown on plain GMM solid agar medium, GMM containing 25 μg/mL Congo Red, and GMM containing 0.05, 0.25, 1, and 8 μg/mL caspofungin. In all, 2000 spores were inoculated and grown at 37 °C for 48 h before visual examination.

**Statistical analysis**. All data are presented as mean value ± standard error, calculated from three or more independent replicates. For comparisons between two treatment groups, the Welch's two-sided t test was performed. Comparisons across multiple conditions or strains were performed using Brown–Forsythe and Welch ANOVA tests followed by Dunnett's T3 multiple comparison test. Comparisons of two strains with two or more conditions were performed via two-way ANOVA test followed the Holm–Šídák multiple comparison test to compare between strains within each treatment. Comparisons of multiple strains grown in the same condition were made using one-way ANOVA followed by Dunnett's multiple comparison tests. All analyses used a cut-off P value = 0.05 for statistical significance.

**Reporting summary**. Further information on research design is available in the Nature Research Reporting Summary linked to this article.

## Data availability

RNA sequencing data supporting the findings in this study have been deposited to the NCBI Gene Expression Omnibus with the identifier GSE156537. All data obtained to support the findings of this study are available within the article and its supplementary materials, or from the corresponding author upon request. Source data are provided with this paper.

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

## Acknowledgements

This study was funded in part by the National Institutes of Health R01 AI065728-01 and GM112739-02 to N.P.K., a Predoctoral Training Program in Genetics award for GF (5T32GM007133), NIH T32 ES007015 to B.N.S., and a Wellcome Trust grant 208396/Z/17/Z to M.B.. We thank Dr. Matthew Henke and Dr. Neil Kelleher at the Northwestern University for advising on tandem mass spectrometry based analysis of oxylipins. Lastly, we thank Dr. Deok-Kun Oh for generously gifting the purified 5,8-diHODE used in this study.

## Author contributions

M.N., B.N.S., G.J.F., N.T.V., N.L.R., and N.P.K. developed the study and designed the experiments. M.N., B.N.S., G.J.F., N.T.V., N.L.R., M.A.W., J.W.B., C.G., C.Z., and E.B. performed the research. M.N., B.N.S., G.J.F., N.T.V., N.L.R., M.A.W., and N.P.K. analyzed and interpreted the data. M.N., G.J.F., and N.P.K. wrote the paper. E.O., D.B., M.B., and N.P.K. provided supplies for the study. N.P.K. is the corresponding author of the paper. All authors reviewed the paper.

## Competing interests

The authors declare no competing interests.
