## [Peer Review File · Nature Communications]

REVIEWERS' COMMENTS:

Reviewer #1 (Remarks to the Author):

The authors have responded to my first set of comments appropriately and thoughtfully. The manuscript has been improved significantly and in my opinion, is suitable for publication in its current form.

Reviewer #3 (Remarks to the Author):

I reviewed the prior version of the manuscript, and noted at that time that the results represent a significant new advance in our ability to understand the regulation of hyphal branching in fungi. I believe that this is still the case, and that the authors have satisfactorily addressed the concerns that I previously raised. The additional data on the phenotypes caused by ZpfA over-expression, and the linkage of GPCRs to the 5,8-diHODE response are compelling and strengthen the authors conclusions.

I have the following minor suggestions for the authors to consider;

line 91 -- please rephrase "...we hypothesized that the lack of metabolite..."

line 168 -- other than the oxylipin phenotype, is there anything else known about gprG in other *Aspergilla*?

lines 212 and 215 -- please check the accuracy of these references. Is AsIA described in ref. 30?

line 227 -- it would be helpful if the images provided in Figs. 6a and 6b were enlarged.

REVIEWERS' COMMENTS:

Reviewer #1 (Remarks to the Author):

The authors have responded to my first set of comments appropriately and thoughtfully. The manuscript has been improved significantly and in my opinion, is suitable for publication in its current form.

Response: We thank the reviewer their remarks.

Reviewer #3 (Remarks to the Author):

I reviewed the prior version of the manuscript, and noted at that time that the results represent a significant new advance in our ability to understand the regulation of hyphal branching in fungi. I believe that this is still the case, and that the authors have satisfactorily addressed the concerns that I previously raised. The additional data on the phenotypes caused by ZpfA over-expression, and the linkage of GPCRs to the 5,8-diHODE response are compelling and strengthen the authors conclusions.

Response: We thank the reviewer their remarks.

I have the following minor suggestions for the authors to consider;

line 91 -- please rephrase "...we hypothesized that the lack of metabolite..."

Response: Revised.

line 168 -- other than the oxylipin phenotype, is there anything else known about gprG in other *Aspergilla*?

Response: At this point, there are no papers focused on GprG in other *Aspergilli* although we would be interested to look more closely at this receptor in both *A. fumigatus* and *A. nidulans* in the future.

lines 212 and 215 -- please check the accuracy of these references. Is AsIA described in ref. 30?

Response: Thank you for finding this, we have fixed and double checked all references.

line 227 -- it would be helpful if the images provided in Figs. 6a and 6b were enlarged.

Response: Fig S6a is a heatmap, and S6b is list of GO terms and part of the Supplemental Data. The images are a function of the NC website and we think the best solution is for a reader to download and enlarge on their computer screen.